# Dog Owners' Survey reveals Medical Alert Dogs can alert to multiple conditions and multiple people

Catherine Reeve[1]*, Clara Wilson[1], Donncha Hanna[1], Simon Gadbois[2]

**1** The School of Psychology, Queen's University Belfast, Belfast, Northern Ireland, **2** Department of Psychology and Neuroscience, Life Sciences Centre, Dalhousie University, Halifax, Canada

* c.reeve@qub.ac.uk

**Data Availability Statement:** All relevant data are within the paper and its Supporting Information files.

## Abstract

Medical Alert Dogs (MADs) are a promising support system for a variety of medical conditions. Emerging anecdotal reports suggest that dogs may alert to additional health conditions and different people other than those that they were trained for or initially began alerting. As the use of medical alert dogs increases, it is imperative that such claims are documented empirically. The overall aims of this study were to record the proportion of MAD owners who have a dog that alerts to multiple health conditions or to people other than the target person and to determine whether any sociodemographic variables were associated with dogs alerting to multiple conditions, multiple people, or both. MAD owners completed an online survey that contained a series of forced choice questions. Sixty-one participants reported a total of 33 different conditions to which dogs alerted. Eighty-four percent of participants reported that their dog alerted to multiple conditions and 54% reported that their dog alerted to multiple people. This is the first study to document that a large percentage of people report that their MAD alerts to multiple conditions and/or to multiple people. We present a discussion of how these alerting abilities could develop, but questions about the underlying mechanisms remain.

## Introduction

Over the past several decades, mounting evidence has emerged for the health and well-being benefits associated with dog ownership [1–3] although this has not been universally accepted [4]. Benefits of dog ownership may manifest in several ways, including decreasing loneliness [5], providing emotional support through physical or psychologically difficult times (e.g., [6–8]), increasing the number of social contacts [9] and increasing activity levels as well as overall health (e.g. [10]). One way in which dogs have the potential to substantially impact their owners' health is by alerting their owner to physiological changes. This is when a dog detects changes in the owner's physiology and the owner is made aware of this through changes in their dog's behaviour. Dog owners have reported behaviour changes in their dogs prior to migraines [11], decreases in blood sugar levels [12,13], the onset of seizures [14,15], and behaviours directed towards areas affected with cancer ([16] c.f. [17]). In these instances, owners

**Funding:** The author(s) received no specific funding for this work.

also report changes in their dogs' behaviour even before they recognise their own associated symptoms. For example, Williams and Pembroke [16] were the first to report on a woman whose dog showed persistent interest in a skin lesion that later turned out to be malignant melanoma, and in a study conducted by Wells et al. [13], 33.6% of dog owners with diabetes reported changes in their dogs' behaviour before they themselves recognised their own symptoms associated with hypoglycaemia. Taken together, these findings suggest that dogs are detecting something not yet perceptible to the person [13,18].

Given dogs' olfactory acuity [19–21], it is postulated that dogs are detecting volatile organic compounds (VOCs) associated with changes in their owners' physiology [22]. VOCs are chemical compounds that have a high vapour pressure and thus exist in gaseous form at room temperature. Within the human body, endogenous VOCs are produced during metabolic processes of cells, are released through breath and are present in the headspace of sweat, urine, faeces, and blood [23]. Analytical chemistry has revealed that specific patterns of VOCs have the potential to serve as odour biomarkers for metabolic conditions and disease states [24], including, but not limited to, cancers [25], diabetic hyper- [24] and hypoglycaemia [26], asthma [27], and epileptic seizures [28]. Empirical studies with olfactory detection dogs support these findings; dogs have been shown to detect odours associated with many of the same conditions including, cancers [29], epileptic seizures [30], and hypo- and hyperglycaemia [31,32].

Following such reports, charities and training organizations have begun to harness dogs' ability to detect odour cues associated with physiological changes in humans. Medical Alert Dogs (MADs) are now trained and placed in homes to alert people to a range of health conditions, including diabetes (hypo- and hyperglycaemic episodes; [12,33,34], epileptic seizures [35], asthma attacks [36], allergic reactions [37,38], Addison's disease [39,40] and Postural Orthostatic Tachycardia Syndrome (POTS) episodes [41]. Specific training protocols vary across charities, training establishments and owners who train their dogs themselves. Despite this, the basic protocol is usually a version of teaching the dog their 'target odour' (e.g., breath or sweat samples collected when that person experiences the medical condition that the dog is intended to alert to) followed by the shaping of 'alerting' behaviours appropriate to communicate to the owner the presence of the target odour. It is important to note, however, that dogs may employ a range of behaviours to communicate an alert, even if they have undergone the same training protocols [34]. Some owners may prefer specific 'alert' behaviours such as picking up a bringsel (a short item suspended from the collar of a detection dog that the dog takes in its mouth as an alert behaviour) or fetching a medical bag, whereas others may utilise behaviours such as the dog staring, nuzzling, or pawing. Some MADs receive no formal training and the dog seemingly spontaneously begins to 'alert' during, or immediately prior to, episodes of their owner's medical condition (e.g., [34]). In this instance, owners may establish these behaviours, through reinforcement, over repeated exposures. It is possible that certain aspects of the dogs' training, for example where they were trained, or how their alerts are responded to, could predict certain behavioural outcomes.

Despite the increased popularity of MADs, a detailed understanding of exactly what a dog is responding to, and the full impact of the owner-dog training interaction, is not yet fully understood. A noteworthy phenomenon is emerging whereby MAD owners report that their dog alerts to health conditions outside of what they were originally trained to alert, or first began alerting. For example, many dogs trained to alert to hypoglycaemia begin spontaneously alerting their owner to hyperglycaemia [34,42]. Furthermore, dogs trained to alert to hypoglycaemia have been reported to alert both their owner and/or other people to other health conditions such as anxiety or asthma attacks (Olivia Rockaway, February 2020). This phenomenon has yet to be documented empirically and is thus the focus of the current study.

Therefore, using an online survey we conducted an exploratory study on a sample of MAD owners. The aims of this study were to document sociodemographic information for MADs and their owners, to document the proportion of MAD owners that report that their dog alerts to multiple conditions and/or multiple people, and to determine whether any sociodemographic variables were associated with whether or not a dog alerted to multiple conditions, multiple people, or both.

## Materials and methods

### Study design

This study utilised a cross-sectional, retrospective design. Participants completed an online survey (described below) that was prepared and presented to participants through the online survey platform, Qualtrics.

### Participants and recruitment

**Recruitment.** Participants were recruited worldwide through advertisements on social media websites (e.g., Facebook, Instagram, and Twitter) related to dog ownership, service dogs, and relevant medical conditions. Responses were collected from May 2019 until August 2019.

**Inclusion criteria.** Prospective participants for this study were required to be at least 16 years of age, be able to complete the survey in English, have access to a device to complete the study online, and have a dog that alerted themselves, or someone they cared for, to a medical condition. The dog could be trained specifically for medical alert or could have started alerting without any formal training.

### Survey

A bespoke survey was designed to assess owner and dog sociodemographic variables, as well as the conditions and people to whom the dogs alerted (available upon request to the corresponding author). The survey consisted of three sections of questions. In the first section, participants were asked about the nature of their relationship with the alerting dog (do they: own a trained medical alert dog that alerts to themselves, own a dog that alerts to themselves without formal training, care for a person that has a trained medical alert dog, or care for a person who has a dog that alerts a medical condition without any formal training), and if the dog was formally trained, who trained the dog. Participants were also asked if their dog receives maintenance training to continue the alerting behaviour, and if so, who completes the maintenance training (a charity or organisation trainer, a private trainer, themselves). These questions were followed by sociodemographic questions including the gender and age of the person to whom the dog alerted, the age, sex, and breed of the dog, how long the dog and the target person had lived together, the relationship between the target person and the dog (the target person does not like their dog, the target person is indifferent towards their dog, the target person sees the dog as a pet and good companion, the target person and dog are best friends), and how friendly the dog is towards people other than the target person (the dog does not like other people, is indifferent towards other people, likes attention from other people, or loves attention from other people). Participants whose dogs began alerting without any formal training were asked how soon after getting their dog did it begin alerting to a medical condition (less than six months, six months to a year, one to three years, three to five years, or greater than five years). Participants who were completing the survey for a dog formally trained for medical

alert were asked whether the dog was trained for any other specialised professions (e.g., visual guide dog, hearing dog, police, search and rescue).

The second section of the survey consisted of questions about the condition(s) to which the dog alerted, and how accurate the participant felt the dog was at alerting to the conditions. In this section, participants were asked to identify the first condition to which the dog was trained to alert, or first began alerting. Participants were presented with a matrix question that consisted of a list of conditions (seizures, hypoglycaemia, hyperglycaemia, anxiety, allergic reactions, narcolepsy, migraines, asthma, Addison's disease, and POTS), as well as an option to enter a different condition (if not listed), alongside categories of ratings (0–25%, 25–49%, 50–74%, 75–100%) for participants to identify how accurate they felt the dog alerted to a particular condition.

The third section of the survey asked participants whether the dog alerted people other than the person for whom the dog was trained to alert (or first began alerting to), and to what condition(s) the dog alerted other people. Participants that indicated that their dog alerted to people other than the target person were presented with a matrix question that consisted of a list of people [family member(s) in the same home, family member(s) in a different home, friend (s), stranger(s)] as well as 'other' with the option of entering text, alongside a number rating of how many of each kind of person the dog alerts (one, two, three, and four or more). In a subsequent question, participants were asked, to the best of their knowledge for what condition(s) their dog had alerted these other people (presented as a list the same as the above-mentioned list of conditions, including an option to add different conditions).

## Procedure

Participants completed the survey on a personal electronic device that had access to the internet. Prospective participants followed a link to the survey page where they were first presented with the participant information sheet that provided information regarding the aims of the study, what kinds of questions participants would be asked to answer, and how their data would be used and stored. Participants were then presented with the informed consent form. Participants were required to provide consent by indicating that they agreed to a series of statements prior to starting the survey. Participants were advised that they were welcome to withdraw their participation at any time by closing their internet browser.

## Ethical approval

The study protocol was approved by the Faculty Research Ethics Committee at Queen's University Belfast (EPS 19_98), and the Research Ethics Board at Dalhousie University, Canada (REB 2019–4803).

## Data analyses

Chi-square tests were used to determine whether any significant associations existed between the variables listed in Table 1 and whether or not a dog alerted to multiple conditions, multiple people, and both multiple conditions and multiple people. Fisher's exact tests were used for 2x2 tables due to the small cell sizes, and for data that violated the assumption that the cell is expected to be a value of 5 or more in at least 80% of the cells, and/or that no cell had an expected value of less than one [43]. Due to the high number of comparisons, Šidák corrections were used to avoid false positives in each family-wise set of comparisons. All data were analysed using IBM SPSS Statistics (Version 24).

**Table 1. Demographic variables of Medical Alert Dogs and the target person to whom the dog alerts.**

| All Dogs | n | Freq | % |
|---|---|---|---|
| Nature of relationship with dog | 64 | | |
| Own a trained MAD that alerts to self | | 31 | 49% |
| Care for a person that is partnered with a trained MAD | | 2 | 3% |
| Own a dog that alerts to self without formal training | | 27 | 42% |
| Care for a person that is partnered with a dog that alerts without formal training | | 4 | 6% |
| Is the dog formally trained for medical alert | 64 | | |
| Yes | | 33 | 52% |
| No | | 31 | 48% |
| Who trained the dog | 33 | | |
| Charity | | 8 | 24% |
| Private Trainer | | 4 | 12% |
| Self | | 21 | 64% |
| Does the dog receive maintenance training | 60 | | |
| Yes | | 31 | 52% |
| No | | 29 | 48% |
| Who does the maintenance training | 31 | | |
| A charity/organisation trainer | | 3 | 10% |
| A private trainer | | 4 | 13% |
| Self | | 24 | 77% |
| Gender of target person to whom dog alerts | 64 | | |
| Male | | 10 | 16% |
| Female | | 54 | 84% |
| Age of target person to whom dog alerts | 64 | | |
| Child (5–14) | | 2 | 3% |
| Youth (15–25) | | 14 | 22% |
| Adult (25–64) | | 43 | 67% |
| Senior (65+) | | 5 | 8% |
| Sex of dog | 64 | | |
| Male | | 33 | 52% |
| Female | | 31 | 48% |
| Dog purebred or mixed breed | 64 | | |
| Purebred | | 49 | 77% |
| Mixed breed | | 15 | 23% |
| How long the target person has been paired with the dog | 64 | | |
| Under 1 year | | 9 | 14% |
| 1–3 years | | 29 | 45% |
| 4–6 years | | 11 | 17% |
| 7–10 years | | 10 | 15% |
| >10 years | | 5 | 8% |
| Target person's feelings towards dog | 64 | | |
| Does not like the dog | | 0 | 0 |
| Indifferent towards the dog | | 0 | 0 |
| The dog is a pet and good companion | | 12 | 19% |
| The target person and dog are best friends | | 52 | 81% |
| Friendliness of dog towards people other than target person | 64 | | |
| Does not like other people | | 1 | 2% |
| Indifferent towards other people | | 9 | 14% |

(*Continued*)

**Table 1.** (Continued)

| All Dogs | n | Freq | % |
|---|---|---|---|
| Likes attention from other people | | 24 | 37% |
| Loves attention from other people | | 30 | 47% |
| **Dogs with No Formal Training for Medical Alert** | **n** | **Freq** | **%** |
| How much time with dog before dog began alerting | 31 | | |
| <6 months | | 13 | 42% |
| 6 months—1 year | | 8 | 26% |
| 1–3 years | | 3 | 10% |
| 3–5 years | | 5 | 16% |
| >5 years | | 2 | 6% |
| **Dogs Formally Trained for Medical Alert** | | | |
| Is the dog trained for other specialised activities | 33 | | |
| Yes | | 18 | 54% |
| No | | 15 | 45% |

## Results

### Demographics

A total of 72 people consented to participating in the survey, however eight participants withdrew from the study providing no usable data. A further three participants provided demographic data but did not report on the conditions to which or people to whom the dog alerted. Full data sets were provided by 61 participants. A large majority of participants were from North America (72%), followed by the United Kingdom (20%), Europe (6%), and Australia (2%).

Most participants reported on dogs that alerted to themselves (91%). Just over half (52%) of the dogs reported on were trained for medical alert while 48% of dogs started alerting to medical conditions without any prior training. A majority (64%) of dogs trained for medical alert were trained by the owner. Just over half of participants (52%) reported that their dog received training to encourage it to continue alerting to physiological changes, with the majority of the training being conducted by the owners themselves (77%). Most participants (84%) reported on a dog that alerted to a target female person and the majority of respondents (67%) reported that the target person was between the ages of 24 and 64.

Participants reported on 52% male dogs (48% female dogs), 77% of dogs were purebred, and the average age of the dogs was 63 months (SD = ±41, minimum age = 7 months, maximum age = 172 months). Just under half (45%) of participants reported that the dog had been paired with its target person for between one and three years. A large proportion (81%) of participants reported that the dog and the target person were best friends, and the large majority of participants reported that their dog liked (37%) or loved (47%) attention from people. Of those dogs that were trained for medical alert, 54% were also trained for other specialised activities (e.g., guide dog, psychiatric assistance/emotional support, search and rescue). For those dogs that began alerting without any formal training for medical alert (n = 31), 42% of participants reported that the dog began alerting within the first 6 months of having the dog. The total distributions of these demographics can be seen in Table 1.

**Table 2. Conditions to which owners report their dogs alert.**

| Condition | Number of times condition was reported | People that reported this condition that also reported that the dog alerts to multiple conditions | | People that reported this condition that also reported that the dog alerts to multiple people | |
|---|---|---|---|---|---|
| | | n | % | n | % |
| Anxiety | 28 | 26 | 93% | 16 | 57% |
| Hypoglycaemia | 27 | 24 | 89% | 18 | 67% |
| Hyperglycaemia | 20 | 19 | 95% | 12 | 60% |
| Migraine | 19 | 17 | 89% | 12 | 63% |
| Seizure | 11 | 9 | 82% | 4 | 36% |
| POTS* | 11 | 11 | 100% | 5 | 45% |
| Allergic reaction | 4 | 4 | 100% | 3 | 75% |
| Narcolepsy | 4 | 4 | 100% | 2 | 50% |
| Asthma | 3 | 3 | 100% | 3 | 100% |
| Periodic paralysis | 3 | 3 | 100% | 3 | 100% |
| Arthritis | 3 | 3 | 100% | 2 | 67% |
| Cancer | 2 | 2 | 100% | 2 | 100% |
| Dissociative episodes | 2 | 2 | 100% | 2 | 100% |
| Dystonia | 2 | 2 | 100% | 1 | 50% |
| Heart complications | 2 | 2 | 100% | 2 | 100% |
| Muscle spasms | 2 | 2 | 100% | 2 | 100% |
| Addison's Disease | 1 | 1 | 100% | 1 | 100% |
| Ankle sprain | 1 | 1 | 100% | 1 | 100% |
| Blackout | 1 | 1 | 100% | 1 | 100% |
| Cataplexy | 1 | 1 | 100% | 1 | 100% |
| Ehlers-Danlos Syndrome | 1 | 1 | 100% | 1 | 100% |
| Pancreatitis | 1 | 1 | 100% | 1 | 100% |
| High blood pressure | 1 | 1 | 100% | 1 | 100% |
| Knee injury | 1 | 1 | 100% | 1 | 100% |
| Low oxygen | 1 | 1 | 100% | 1 | 100% |
| Sepsis | 1 | 1 | 100% | 1 | 100% |
| Sinus tachycardia | 1 | 1 | 100% | 1 | 100% |
| Depression | 1 | 1 | 100% | 0 | 0 |
| Cluster headache | 1 | 1 | 100% | 0 | 0 |
| Postural hypotension | 1 | 1 | 100% | 0 | 0 |
| PTSD† | 1 | 1 | 100% | 0 | 0 |
| Sleep apnea | 1 | 1 | 100% | 0 | 0 |
| Syncope | 1 | 1 | 100% | 0 | 0 |

*POTS: Postural Orthostatic Tachycardia Syndrome

† Post Traumatic Stress Disorder.

## Conditions to which dogs alert

Participants reported a total of 33 different conditions to which dogs alerted (listed in Table 2). The most common conditions that dogs were reported to alert to were anxiety, hypoglycaemia, hyperglycaemia, migraines, seizures, and POTS. A large majority (84%) of participants reported that their dog alerted to more than one condition (Table 3), with the average number of conditions that dogs were reported to alert to being $M = 2$ (SD = 1.66, Min = 1, Max = 9). Dogs that alerted to anxiety, hypoglycaemia, hyperglycaemia, migraines, seizures, or POTS

**Table 3. The percentage of participants that report that their dog alerts to multiple conditions, multiple people, and both.**

| | | Dog Alerts to Multiple People n (%) | | |
| --- | --- | --- | --- | --- |
| | | **Yes** | **No** | **Total** |
| **Dog Alerts to Multiple Conditions %(n)** | **Yes** | 28 (46%) | 23 (38%) | 51 (84%) |
| | **No** | 5 (8%) | 5 (8%) | 10 (16%) |
| **Total** | | 33 (54%) | 28 (46%) | 61 (100%) |

were also frequently reported as alerting to other conditions, however, Fishers exact tests with a Šidák correction for multiple comparisons (original $\alpha$ = .05 $\alpha_{SID}$ = .009) revealed that alerting to any of these six conditions was not significantly associated with whether or not a dog alerted to multiple conditions (anxiety $p$ = .092, hypoglycaemia $p$ = .489, hyperglycaemia $p$ = .144, migraine $p$ = .485, seizure $p \approx 1.000$, and POTS $p$ = .184; for those dogs that alerted to anxiety, hypoglycaemia, hyperglycaemia, migraines, seizures, or POTS, the other conditions that they were reported to alert to are listed in the supplementary materials). When asked to report on the accuracy with which dogs alerted to conditions, owners perceived their dogs to be highly accurate, with over 70% of ratings being between 75%-100% (see Table 4).

Fisher's Exact tests were completed for all variables in Table 1 (except the nature of the relationship with the alerting dog) and whether or not a dog alerted to multiple conditions. Analyses revealed that, when a Šidák correction for multiple comparisons was applied (original $\alpha$ = .05, $\alpha_{SID}$ = .004), for dogs without formal training for medical alert, the amount of time the primary person had been with their dog before it began alerting was marginally significantly related to whether or not the dog alerted to multiple conditions (see supplementary materials). The highest proportion of dogs that alerted to multiple conditions alerted within the first 6 months, 42% (13/31), followed by 26% (8/31) between 6 months to a year, 16% (5/31) 3 to 5 years, 10% (3/10) 1 to 3 years, and 6% (2/31) greater than 5 years (see Table 5) ($p$ = .004, two-sided Fisher's Exact Test). Whether or not the dog was formally trained for medical alert, who trained the dog, whether or not the dog received maintenance training, and who conducted the maintenance training were not significantly associated with whether or not the dog alerted to multiple conditions. Furthermore, the age of the target person to whom the dog alerted, the sex of the dog, how long the target person had been with their dog, the target person's feelings towards the dog, the friendliness of the dog towards people other than the target person, were not significantly associated with whether or not a dog alerted to multiple conditions. For those dogs that were formally trained for medical alert, whether or not the dog was trained for other specialised activities were not significantly associated with whether or not the dog alerted to multiple conditions (see supplementary materials).

## People to whom dogs alert

Just over half of participants (54%) reported that their dog alerted medical conditions to people other than the target person (see Table 3). Dogs were reported to alert to family members in

**Table 4. Accuracy with which dogs are reported to alert to condition(s).**

| Accuracy Rating | Dog alerts to multiple conditions n (%) | Dog alerts to a single condition n (%) |
| --- | --- | --- |
| 0–24% | 8 (6.2%) | 0 |
| 25–49% | 13 (10%) | 0 |
| 50–74% | 11 (8.5%) | 1 (10%) |
| 75–100% | 98 (75%) | 9 (90%) |

**Table 5. The amount of time a dog with no formal training for medical alert spent with a target person before the dog began alerting to medical conditions, and whether or not the dog alerts to multiple conditions.**

| | | Amount of time target person spent with dog before dog began alerting n(%) | | | | |
|---|---|---|---|---|---|---|
| | | <6 months | 6 months -1 year | 1–3 years | 3–5 years | >5 years |
| **Does the Dog Alert to Multiple Conditions?** | **Yes** | 12 (39%) | 8 (26%) | 1 (3%) | 4 (13%) | 0 |
| | **No** | 1 (3%) | 0 | 2 (6%) | 1 (3%) | 2 (6%) |

the same home, family members in different homes, friends, and strangers (see Table 6). These were instances of the dog alerting someone (other than their owner) to a health condition specific to that person, rather than the dog alerting another person to get attention for their target person. Considering the top six conditions that dogs were reported to alert to, Fishers exact tests with a Šidák correction for multiple comparisons (original $\alpha$ = .05, $\alpha_{SID}$ = .009) revealed that there were no significant associations between whether or not a dog alerted to these conditions and whether or not a dog alerted to multiple people (anxiety, $p$ = .797, hypoglycaemia, $p$ = .121, hyperglycaemia, $p$ = .591, migraine, $p$ = .412, seizure, $p$ = .317, and POTS, $p$ = .740). The conditions to which dogs were reported to alert to other people and the number of participants that reported their dog alerted other people to those conditions can be seen in Table 7. Of the 33 respondents that reported that their dog alerted to multiple people, only 6% (n = 2) reported that their dog alerted other people to condition(s) other than conditions to which the dog alerted the target individual (see supplementary materials), while 94% of participants reported the dog alerted other people to the same conditions to which it alerts the target individual.

Fisher's Exact Tests revealed no associations between sociodemographic variables and whether or not a dog alerted to multiple people (see supplementary materials).

## Dogs alerting to both other conditions and other people

Nearly half (46%) of participants reported that their dog alerts to both other conditions and other people (Table 3). For each condition reported, the percentage of people that reported that their dog alerted to that condition and also reported that the dog alerted to multiple conditions or multiple people can be seen in Table 2. Considering the top six conditions that dogs were reported to alert to, Fisher's Exact Tests with a Šidák correction for multiple comparisons revealed no significant associations between these conditions and whether or not a dog alerted to both multiple people and multiple conditions (anxiety, $p$ = .612, hypoglycaemia, $p$ = .075, hyperglycaemia, $p$ = .172, migraine, $p$ = .270, seizure, $p$ = .526, and POTS, $p$ = .1.000). Furthermore, additional Fisher's Exact Tests with a Šidák correction for multiple comparisons revealed no associations between the sociodemographic variables and whether or not a dog alerts to both other conditions and to people other than the target person (see supplementary materials).

**Table 6. The number of participants that reported that their dog alerted to 1, 2, 3, or 4 or more people other than the target person.**

| Other people to whom dog alerts | 1 | 2 | 3 | 4 or more |
|---|---|---|---|---|
| | No. of participants | | | |
| **Family member in the same home** | 11 | 4 | 1 | 2 |
| **Family member in a different home** | 4 | 3 | 0 | 4 |
| **Friends** | 5 | 5 | 4 | 5 |
| **Strangers** | 4 | 6 | 0 | 9 |

**Table 7. The conditions to which dogs were reported to alert to other people and the number of participants that reported their dog alerted other people to those conditions.**

| Condition | No. of participants |
|---|---:|
| Hypoglycaemia | 15 |
| Anxiety | 12 |
| Hyperglycaemia | 6 |
| Migraine | 5 |
| Cancer | 2 |
| Seizure | 2 |
| Dissociative episodes | 1 |
| Heart attack | 1 |
| POTS* | 1 |
| Allergic reactions | 1 |
| Low Oxygen | 1 |
| Dystonic spasms | 1 |
| Knee injury | 1 |
| Ankle sprain | 1 |
| Shoulder spasm | 1 |
| Periodic paralysis | 1 |
| Asthma | 1 |
| High blood pressure | 0 |

*POTS: Postural Orthostatic Tachycardia Syndrome.

## Discussion

The aims of this study were to document sociodemographic information for MADs and their owners, to document the proportion of MAD owners that report that their dog alerts to multiple conditions and/or multiple people, and to determine whether any sociodemographic variables were associated with whether or not a dog alerted to multiple conditions, multiple people, or both. Participants completed an online survey that gathered sociodemographic information for the target person and the dog, and had participants report on the conditions to which and people to whom the dog alerts. The main findings were that participants reported a total of 33 different conditions to which their dogs alerted, a large majority of dog owners reported that their dog alerts to multiple conditions (84%), and over half of respondents reported that their dog alerts to multiple people (54%). Just under half of participants (46%) reported that their dog alerts to both other conditions and other people. This is the first study of its kind to document the phenomenon of dogs, both specially trained for medical alert, and dogs with no specific training for medical alert, alerting to multiple different conditions, and to multiple different people.

Although the aim of this study was not to determine how, or why, dogs may alert their owner, and others, to multiple health conditions, we will discuss possible factors that may contribute to this phenomenon and how these factors could explain our findings. First, it could be the case that a similar physiological state precedes a number of the conditions reported and that the dogs are detecting this preceding state. For example, the most commonly reported condition in this study was anxiety and of those who reported that their dog alerted to anxiety, 93% of participants reported that their dog also alerted to other conditions (however this relationship was not significant). Furthermore, although there was no significant association between whether or not a dog alerted to anxiety and whether or not a dog alerted to multiple

people, anxiety was the second most frequently reported condition that dogs alerted to other people. There is a known association between anxiety and epilepsy [44], and there is evidence for a link between anxiety and migraines [45,46]. It is possible that an individual experiences anxiety or stress before a seizure, migraine, or other physiological changes and their dog simply detects the stress preceding any host of physiological changes. Participants also reported that their dogs have alerted to injuries such as sprains. Although a dog can only be aware of an injury after it happens, the physiological and emotional stress experienced by the person along with the injury could signal to the dog that their owner is experiencing a change in overall well-being. While there is some evidence that psychological stress produces detectable VOCs [47] and that dogs can detect odour cues associated with fear [48] and stress [49], the claim that stress precedes multiple conditions requires further investigation.

Next, although analytical chemistry has revealed potential VOC profiles for a wide number of conditions, different analyses of the same condition often yield differing results [50,51] and conversely, analysis of different conditions reveal overlap in VOCs [52]. Analysis of VOCs is complicated by the fact that there is large inter-person variability in VOCs; differences in diet, medications, metabolism, and environmental exposures can result in variability in endogenous and exogenous VOCs both within and between people [53]. Overall, defining the VOC profiles for individual health conditions is still ongoing and there is need for further understanding of how these profiles may be similar or different between individuals and how this may translate to the behaviour of a dog. It is possible that the same, or closely related, VOCs are emitted across different conditions or by different people, and dogs that alert to multiple conditions and/or multiple people are detecting these similarities. In this case, a dog may provide an alerting response because the odour is 'close enough' to the target odour. Whether or not a dog considers an odour to be 'close enough' is the second factor that could explain a dog alerting to multiple conditions or multiple people.

MADs are exposed to a continuous stream of odours at all times. Therefore, the detectability of the VOCs associated with specific physiological changes can be obscured by the background 'noise' of other odours. One component of a dog's ability to detect a specific odour is their biological 'hardware'. Dogs have upwards of 200 million olfactory receptors to which odorants bind and where signal transduction to the brain occurs [54]. However, the exact number of olfactory receptor cells in a dog's nasal cavity is variable and dependent upon dog breed and genetics [55]. Furthermore, the genes that code for these receptors have documented allelic variation [55] and studies suggest that a dog's ability to detect a target odour is related to particular alleles [54]. As such, a dog's ability to detect specific odour(s) associated with a condition and therefore alert to the condition may be, in part, due to their underlying biology.

Combined with their perceptual ability, other factors that could impact whether or not a dog alerts to a medical condition are affective and temperamental differences in emotional states impacting attention and decision making [56–59]. In other words, each dog will have a threshold for whether they perform the alert behaviour to a target odour, or not, when they are uncertain about the choice to make. Signal Detection Theory [60] terms this threshold an individual's 'criterion' and each individual's criterion exists on a continuum from conservative to liberal. Considering MADs, a more conservative dog could be one that only alerts when they are confident that an odour represents the condition that they have been trained to alert to or have been reinforced for alerting to previously. Conservative dogs might be less likely to alert when the condition is not being presented (minimising false alarms) and could be more likely to withhold an alert when the condition is in fact present (thereby committing more misses). Following this logic, conservative dogs may be less likely to alert to multiple conditions or multiple people, as they would be less likely to respond to odours that are similar, but not exact to,

those on which they were trained. On the other hand, a liberal MAD would be more likely to alert to odours that are 'close enough'; odours that only approximate the condition of interest. As such, liberal dogs could be more likely to commit false alarms but less likely to miss instances of a condition. In the context of the current study, a liberal MAD would be more likely to generalise their alert response to odours that are not exact to those to which it had previously been reinforced, therefore being more likely to alert to multiple conditions and/or multiple people. Empirical tests in controlled laboratory settings have allowed researchers to measure the parameters of Signal Detection Theory and have highlighted individual differences in dogs' sensory perceptual abilities and decision criteria during human odour detection tasks [32,61]. Although a dog may have a natural inclination to be a conservative or liberal decision maker, their decision-making bias can be manipulated through training and reinforcement. Therefore, an additional factor that may influence the likelihood that a dog alerts to multiple conditions, or multiple people, is their reinforcement history.

A dog's reinforcement history could play a large role in the likelihood of them demonstrating alerting behaviours to multiple conditions or people. Most MADs are trained using positive reinforcement for a correct alert (e.g., they receive praise or a treat after successfully alerting their owner to a condition). By definition, reinforcing the alert behaviour will increase the likelihood that the behaviour will occur again in the future [62]. A liberal dog may present the alerting behaviour in different contexts and to different odours. If the alert behaviour occurs when the owner or another person is experiencing a physiological change and if the alert behaviour is then reinforced properly, the dog can become conditioned to respond to a different health condition or to a different person. Similarly, if an owner demonstrates particular behaviours associated with the onset of a condition, and these behaviours elicit attention seeking or affiliative behaviours from their dog, the owner may, over time, interpret the dog's behaviours as an alert and reward them accordingly. As such, the owner has again conditioned an alerting response to a new condition. This same process can be applied to dogs being rewarded for trying their alert behaviour directed to another person.

It should be noted that fifty-two percent of owners in the study sample reported that their dog receives ongoing maintenance training to continue alerting. It is important to consider what may constitute 'training'. An owner may have interpreted this term as meaning formal engagement with specific training methods, or hiring a trainer, for example. However, it can be argued that any form of reinforcement following a correct alert is a form of training, in that you are increasing the likelihood of that behaviour occurring in future. Despite only fifty-two percent of owners reporting that their dog receives ongoing training, the vast majority of owners reported that their dog was highly accurate at alerting (75–100% correct). Wilson et al. [34] found that Diabetes Alert Dog owners who were most compliant with their ongoing training protocol (e.g., correctly rewarding true positives and ignoring false positives) had better performing dogs than those owners who did not comply with protocol. It is possible that there was a misinterpretation of the word 'training' in this instance, and that more owners are positively reinforcing their dog's alerts than were reported. Alternatively, or additionally, it is possible that our obtained reports of accuracy are biased by the fact that it is an owner reporting on their own dog. This phenomenon has been established previously within Diabetes Alert Dog owners, where owner reports of accuracy were higher than objective measures of performance (e.g. [33]) The current study includes MADs who, all together, are reported to alert to 33 different health conditions, with many dogs reported to have received no formal training, receive no ongoing training, and yet provide high levels of alerting accuracy. Results such as these highlight the need for objective assessments of MAD performance and behaviour in future studies. However, as this was a preliminary and exploratory study, the possible bias in owner-reporting was accepted as an aspect of documenting this phenomenon.

Analyses revealed that the only marginally significant association was that, for dogs with no previous formal training for medical alert, the amount of time the dog had spent with the target person before it began alerting was significantly related to whether or not the dog alerted to multiple conditions. The distribution of data here would suggest that dogs that began alerting their target person within six months of being together were more likely to alert to multiple conditions than those dogs that spent over six months with their target person before they began alerting. As these dogs received no formal training, they initiated 'alert' behaviours which were likely then developed into consistent alerts through the owner's response to the alerting behaviour over time. Given that this was not a trained response, these dogs likely had a strong natural response to either the owner's odour profile, or behavioural changes associated with a health-related episode, which then resulted in changes in their own behaviour. The dogs' behavioural responses to their owners' physiological changes may have been differentially reinforced for some conditions and not others. There are any number of reasons why this could have occurred, including whether or not the owner actually linked the dogs' behaviour to a specific physiological change, or whether the owner did not care about or need their dog alerting to a specific physiological change and therefore did not reinforce the dogs' behaviour. It could be assumed that this process would take place at the beginning of the human-dog relationship and, once the reinforcement contingencies were in place, the dog learned to discriminate between different conditions and only alert some conditions and not others. Additionally, it is possible that as a dog spends more time with a particular owner, they develop a more thorough understanding of what the owner smells like and what specific odours signify a significantly altered physiological state in that owner.

Analyses revealed that none of the variables assessed were related to whether or not a dog alerted to multiple people. Of interest, however, was that of the thirty-three dogs who were reported to alert to other people, the vast majority of participants reported that dogs alerted the other people to the same conditions to which the dog alerted the target person. This suggests that there is some level of odour consistency for the same condition across different people and is consistent with studies that reveal odour biomarkers associated with specific conditions [24]. It further demonstrates that, within this sample, dogs were unlikely to learn new target odours from a novel person. Thirty-one out of thirty-three dogs were able to generalise odours associated with their owners' condition(s) to a new person but were not seemingly expanding this behaviour to a novel health condition in a novel person. These results highlight the need for future empirical work into the odour profile of certain conditions, and across different individuals, to gain further understanding into what these dogs are detecting, and how this may impact their decision making.

But as reported in the results, two dogs were reported to alert other people to conditions other than those that they alerted the target person. For the first dog, the conditions that the dog was reported to alert to other people were injuries (sprains) and muscle spasms. In the case of a dog alerting to injuries, it is difficult to determine whether the dog is in fact alerting to the injury itself or other physiological states associated with the injury. Since this dog also alerted the target person to anxiety, it is possible that the dog was detecting anxiety associated with the injuries and not the injuries themselves. Given that this was the only dog reported to alert to injuries, it is difficult to discuss the phenomenon further. The second dog, however, was reported to alert other people to two different cardiovascular conditions (heart attack and high blood pressure). Again, this dog alerted the target person to anxiety, so it is possible that anxiety played a role in the dog alerting other people to a heart attack and high blood pressure. Furthermore, it is also possible that, for both dogs, the conditions to which they alerted other people were events that elicited attention seeking behaviours from the dogs, and these behaviours were interpreted as alerts. This interpretation of the findings is most likely given that the

dogs were unlikely to have reinforcement histories associated with the conditions to which they alerted other people.

Although this study was exploratory in nature, the main limitation of this study was the small sample size which limited the power of any analyses. A further limitation of the study is that the reported results may be impacted by the demographics of the sample; the majority of the sample identified as female and were between the ages of 24–65. Therefore, more generalized comments across MAD owners as a whole cannot be made until further cohort studies have been carried out. In addition, those that responded to the survey may feel more positively about their MAD than those who did not take part in the survey. Indeed, of those that did respond, 81% classified themselves and their MAD as 'best friends'. It should also be noted that the study was a self-report survey which means that MAD owners reported on their *perception* of their dogs' alerting behaviours. Therefore, it is possible that individual and cultural differences in interpreting dogs' behaviour could have affected the owners' responses [63] As such, the subjective nature of the study means the findings should be interpreted accordingly. A lack of objectivity likely influences owners' reports of what their dogs alert to and their perceived level of their dogs' accuracy [64]. Moreover, the study information sheet presented to participants before they consented to taking part in the study explicitly stated that the researchers were interested in how common it was for dogs that alert to medical conditions to alert to more than one condition or person. Although we stated that we also wanted to hear from dog owners whose dogs did not alert to multiple conditions or people, it is possible that demand characteristics resulted in participants unconsciously misreporting their dogs' alerting behaviours. While the current study documents a previously anecdotal phenomenon, follow up objective studies are needed to assess the frequency and development of these behaviours. Our results should be interpreted cautiously.

Almost half of respondents stated that their dog had received no formal training for medical alert, and of those that were trained, seventy-seven percent were trained by the owners themselves. It is possible that dogs who have had formal training (e.g., from an accredited training establishment or charity) for a single odour and have an owner that is less likely to reward any alerts outside of the specific criteria, are less likely to alert to multiple conditions or multiple people. However, within this sample, whether the dog was formally trained or not was not significantly associated with whether or not the dog alerted to multiple conditions or multiple people. It should be noted, however, that dogs formally trained from training establishments/charities represented only twenty-three percent of the sample, therefore any differences may only emerge with more participants per group.

This study sought to capture a sample representative of MADs and, as many operational MADs are working without having formal training, it was considered important that all MAD owners were included in the sample. Overall, given the small number of participants representing certain training categories, results pertaining to factors that may impact the likelihood that a dog may, or may not, alert to multiple conditions or people should be taken with caution. However, despite the small sample size, it is clear that many MAD owners report that their dog alerts to multiple conditions, people, or both and that MAD owners perceive their dog to be highly accurate in their alerts.

## Conclusion

This study sought to document the phenomenon that dogs alert to multiple health conditions and to multiple people. The results showed that a large majority of MAD owners reported that their dog alerts to multiple conditions, over half of respondents reported that their MAD alerts to multiple people, and just under half of participants reported that their dog alerted to both

multiple conditions and multiple people. Dogs were reported to successfully alert to 33 different health conditions, and mostly alerted other people to the same conditions that they alerted the target person. Owners perceived their dogs to be highly accurate with their alerts. The results may suggest overlapping VOC profiles between different conditions, as dogs alerting to multiple health conditions and multiple people could suggest that there are some common odours across conditions to which the dogs are responding. Furthermore, it is possible that the phenomena of dogs' alerting to multiple conditions and/or multiple people are, through a combination of the dogs' liberal alerting and the owners' reward contingencies, shaped over time with the owner. Given the results of other studies which found discrepancies between owner reports and objective assessments of MADs, these phenomena should be assessed directly in future studies. Such studies may wish to objectively investigate the rate at which this phenomenon is occurring, and begin to address how, and why, it is emerging and/or reinforced in working MADs. What is apparent from the results of this study is that many untrained and formally trained dogs are reported to be alerting to multiple health conditions and to multiple people.

## Supporting information

**S1 Table. The top six most frequently reported conditions dogs alert to and for dogs that alert to those conditions, the other conditions they are also reported to alert.**
(DOCX)

**S2 Table. The results of Fishers exact tests for sociodemographic variables of the target person and dog and whether or not the dog alerted to multiple conditions, multiple people, or both.**
(DOCX)

**S3 Table. The conditions that dogs alerted other people that were different from the conditions to which the dog alerted the target person.**
(DOCX)

**S1 File.**
(DOCX)

**S1 Dataset.**
(SAV)

## Acknowledgments

Thanks to Leah Cohen for assisting in downloading and coding survey responses.

## Author Contributions

**Conceptualization:** Catherine Reeve, Simon Gadbois.

**Data curation:** Catherine Reeve, Clara Wilson, Donncha Hanna.

**Formal analysis:** Catherine Reeve, Clara Wilson, Donncha Hanna.

**Investigation:** Catherine Reeve.

**Methodology:** Catherine Reeve.

**Writing – original draft:** Catherine Reeve, Clara Wilson.

**Writing – review & editing:** Catherine Reeve, Clara Wilson, Simon Gadbois.

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
