## [Decision Letter · Decision Letter 0]

10 Dec 2020

PONE-D-20-32952

Medical Alert Dogs are alerting to multiple conditions and multiple people

PLOS ONE

Dear Dr. Reeve,

Thank you for submitting your manuscript to PLOS ONE. After careful consideration, we feel that it has merit but does not fully meet PLOS ONE’s publication criteria as it currently stands. Therefore, we invite you to submit a revised version of the manuscript that addresses the points raised during the review process.

You will find the detailed reviewer response below. They draw special attention to the statistical analysis and how it is reported in the paper. In the interest of getting you a timely response to your submission, we are not seeking additional statistical reviewer input on this version, but it is likely that we will do so when sending a revised version for review.

We look forward to receiving your revised manuscript.

Kind regards,

I Anna S Olsson, Ph.D.

Academic Editor

PLOS ONE

Journal Requirements:

2. In your Methods section, please provide additional information about the participant recruitment method and the demographic details of your participants. Please ensure you have provided sufficient details to replicate the analyses such as: a) the recruitment date range (month and year), b) a description of any inclusion/exclusion criteria that were applied to participant recruitment, c) a table of relevant demographic details, d) a description of how participants were recruited, and e) descriptions of where participants were recruited and where the research took place.

3. Please include additional information regarding the survey or questionnaire used in the study and ensure that you have provided sufficient details that others could replicate the analyses. For instance, if you developed a questionnaire as part of this study and it is not under a copyright more restrictive than CC-BY, please include a copy as Supporting Information.

Reviewers' comments:

Reviewer's Responses to Questions

**Comments to the Author**

1. Is the manuscript technically sound, and do the data support the conclusions?

Reviewer #1: Partly

2. Has the statistical analysis been performed appropriately and rigorously? 

Reviewer #1: No

3. Have the authors made all data underlying the findings in their manuscript fully available?

Reviewer #1: Yes

4. Is the manuscript presented in an intelligible fashion and written in standard English?

Reviewer #1: Yes

5. Review Comments to the Author

Reviewer #1: This paper reports on an exploratory study about Medical Alerting Dogs and their owners, about whether the dogs alert to multiple conditions and to multiple owners. It is an interesting and timely piece of work, that is well written and easy to follow.

I do have a few major concerns:

- The authors document sociodemographic information for MAD, but why did they choose not to examine whether people other than the primary person, if dogs alerted to multiple people/multiple conditions, actually had a condition and which one that was?

- When dogs also alerted to other conditions, it would be interesting to examine which these conditions were. Given that there many conditions that were reported infrequently, I would focus this analysis on the more common ones (anxiety, hypoglycaemia, hyperglycaemia, seizure and POTS). This could be a descriptive analysis, not an inferential one, but it would allow to remove some speculation that currently occurs in the discussion.

- The paper does not report any test statistics, degrees of freedom, or P-values. I would like to see these added to substantiate the claims made by the authors.

- In the statistical analysis, many comparisons have been made. Because of this, significant findings can occur by chance. A more conservative way is to correct for the number of statistical tests done by dividing the cut-off value (P-value) by the number of tests run. The authors did not choose to do this, but it should be revised by a statistician whether it is not more correct to do so still.

There are also some minor concerns:

INTRODUCTION

- Line 46: reference 1 is incomplete in the list of references. Please correct.

- Line 54: in addition to reference 14, there is also the work by Martos Martinez-Caja (2019 in Epilepsy & Behavior), which is more recent, to support this.

- Line 88: I do not know the word ‘brinsel’. I have looked it up, but could not retrieve its meaning. Can this word be substituted, if not explained?

- Line 93: “It is possible that certain aspects of their training”. Replace “their” by “the dogs’ “, so that it is more clear.

- Line 107: typo – “on” should be “an”

MATERIALS AND METHODS

- Lines 120-125 and line 128: in which language was the survey designed? For a UK target audience? Is anything known about the country of origin of the participants?

- Line 129-130: rather than having people ask for the survey, I would suggest to include it in full as supplementary material

- Line 167: To make this sentence easier to read, I would add “for” between “knowledge,” and “what condition(s)”

- Line 187: typo – “altered” should be “alerted”

RESULTS

- Table 2: this is a very important table, that could be even more informative if it was also added how many dogs were trained/untrained when they alert to a particular condition. Also, a table or figure should be able to stand on its own, meaning everything in it must be clear. Therefore, I would suggest explaining the abbreviations POTS and PTSD below the table.

- Line 248: the numbers here (90% and 10%) do not correspond to those in table 5. Please check and adjust as necessary.

- Lines 279-281: I would break this down for the most commonly alerted conditions as well.

DISCUSSION

- Line 316 and further: this discussion could be substantiated if the authors examined their data in a bit more detail for the most commonly reported conditions that dogs alert to (see major comment). At the moment it seems that dogs mainly alert the same condition in other people in the household, but surely that is more common for some conditions than others, since it is unlikely e.g. that two people from the same household have epilepsy. Although I do appreciate the conciseness with which the authors present their results, a more detailed presentation of the data would help here.

- Line 360: “for whether they decide to alert” – I would be careful with this kind of working. A dog does not decide to intentionally alert a person. The dog may decide to intentionally perform a particular behavior (which the owner then interprets as an alert).

- Lines 476-477: also, how was the study presented to the respondents? Is there a risk of response bias in that repondents may have felt they were expected to have a dog that responds to multiple conditions/people. Perhaps the authors could elaborate on this as well?

6. PLOS authors have the option to publish the peer review history of their article (what does this mean?). If published, this will include your full peer review and any attached files.

Reviewer #1: No

---

## [Author Response · Author response to Decision Letter 0]

15 Jan 2021

Dear Dr Olsson,

Thank you for taking the time to review our manuscript and have a reviewer provide feedback. We are pleased to have an opportunity to revise our manuscript. In the revised manuscript, we have carefully considered the reviewers comments and suggestions and have addressed and/or responded to each of their points. The reviewer’s comments were very helpful overall, and we are appreciative of such constructive feedback on our original submission. After addressing the issues raised, we feel the quality of the paper is much improved. Following the reviewer’s suggestions, we have added three additional tables as supplementary materials (file entitled: S1 S2 S3 Tables), the captions for which can be found at the very end of the manuscript file. 

Upon closer inspection of the manuscript, we have also made a few further edits (i.e., editing references that were formatted incorrectly). All changes are indicated with track changes in the file named Reeve et al – Revised Manuscript with Track Changes.doc.

Thank you for your time and we look forward to hearing from you.

Sincerely,

Dr Catherine Reeve

Response to Reviewer Comments

Dear Reviewer,

Thank you for taking the time to review our manuscript, Medical alert dogs are alerting to multiple conditions and multiple people. We very much appreciate your feedback. You highlighted a number of ways in which we could improve the quality of our manuscript, and we have made the changes accordingly. Please find our responses to your comments below:

Major Comments

Comment: 

The authors document sociodemographic information for MAD, but why did they choose not to examine whether people other than the primary person, if dogs alerted to multiple people/multiple conditions, actually had a condition and which one that was?

Response:

Thank you for your query. We would like to point out that in lines 169-172 (in the revised manuscript without track changes), participants were asked, to the best of their knowledge, what conditions their dog had alerted to other people. We did not ask participants to report specifically what conditions the dogs were alerting to specific people and therefore cannot report this level of detail. But we agree that the findings related to the conditions that dogs alert other people to were not discussed in much detail and, therefore, the conditions to which dogs were reported to alert to other people have now been presented in Table 7. 

Furthermore, in lines 291-295 we state that 94% of participants reported that if the dog was alerting to other people, it was alerting the same condition(s) to which the dog alerted the target person. Only 6% (n=2) of those who reported that their dog alerted to other people reported that their dog alerted other people to a different condition. In response to your comment, we have added the details of those who reported that their dog alerted other people to conditions other than those conditions to which the dog alerted the primary person in a table in the supplementary materials (S3) and have discussed these findings further in the discussion (Lines 483-497). 

Comment:

When dogs also alerted to other conditions, it would be interesting to examine which these conditions were. Given that there many conditions that were reported infrequently, I would focus this analysis on the more common ones (anxiety, hypoglycaemia, hyperglycaemia, seizure and POTS). This could be a descriptive analysis, not an inferential one, but it would allow to remove some speculation that currently occurs in the discussion.

Response:

You have raised an important point and we agree that knowing which other conditions dogs alert to would be a valuable addition to the manuscript. A supplementary table (S1 Table) has been added that presents the top six most frequently reported conditions that dogs alert to and, for dogs that alert to those conditions, the other conditions they are reported to alert. We have also conducted further analyses to examine whether there are any significant associations between whether or not a dog alerts to each of these conditions and whether or not the dog alerts to multiple conditions (see response to next comment). 

Comment:

The paper does not report any test statistics, degrees of freedom, or P-values. I would like to see these added to substantiate the claims made by the authors.

Response:

Thank you for raising an important point. The only inferential statistics performed were Fisher’s exact tests because comparisons were either 2x2 tables, or, for tables larger than 2x2, one or more of the cells in the table contained counts less than 5 (based on the assumptions of Chi-Square tests of independence as discussed by McHugh, 2013). When using Fisher’s exact tests only the p value is reported.

The authors had previously discussed included binomial logistic regression to analyse whether any of the variables in Table 1 predicted whether or not a dog alerted to multiple conditions, multiple people, or both multiple conditions and multiple people, but considering that a basic assumption of regression analyses is that there are roughly 12 participants per independent variables, our small sample size led us to decide that the results of such analyses would be not be valid. 

But considering the comments about presenting more detailed analyses of the top six conditions to which dogs were reported to alert, additional analyses were included. We analysed whether alerting to one of the top six conditions was associated with a greater likelihood of alerting to other conditions, other people, or both other conditions and other people. Results revealed that there were no significant relationships between the variables. These results are now reported on lines 238-242, 285-289, and 311-314. 

Comment:

In the statistical analysis, many comparisons have been made. Because of this, significant findings can occur by chance. A more conservative way is to correct for the number of statistical tests done by dividing the cut-off value (P-value) by the number of tests run. The authors did not choose to do this, but it should be revised by a statistician whether it is not more correct to do so still.

Response:

Thank you for highlighting this error on our part. We have adjusted our analyses and used a Šidák correction for multiple comparisons. As a result of the correction for multiple comparisons, one of our previously significant findings is no longer significant (association between gender of target person and whether or not the dog alerts to multiple conditions) and as such, discussion of this finding has been removed.

Minor Comments

Introduction

• Line 46: reference 1 is incomplete in the list of references. Please correct.

>> Reference 1 (now on line 558) has been completed.

• Line 54: in addition to reference 14, there is also the work by Martos Martinez-Caja (2019 in Epilepsy & Behavior), which is more recent, to support this.

>> Thank you very much for pointing out this relevant paper. We have included the paper as a citation on line 54 and it is included in the reference list (number 15, line 595).

• Line 88: I do not know the word ‘brinsel’. I have looked it up, but could not retrieve its meaning. Can this word be substituted, if not explained?

>> We appreciate that this term is not one that is commonly used outside of medical detection dog training. We have also identified that it was, in fact, spelled incorrectly. We have therefore edited the spelling of the word accordingly, and have further included a brief definition on lines 89-90.

• Line 93: “It is possible that certain aspects of their training”. Replace “their” by “the dogs’ “, so that it is more clear.

>> The recommended edit has been made on line 94.

• Line 107: typo – “on” should be “an”

>> This error has been corrected on line 108.

Materials and Methods

• Lines 120-125 and line 128: in which language was the survey designed? For a UK target audience? Is anything known about the country of origin of the participants?

>> Thank you for pointing out the lack of detail regarding the survey itself and the participants. The survey was only available in English and available to participants worldwide. We have added this information as well as general descriptive analyses of the locations of participants on lines 122, 126. 203-205.

• Line 129-130: rather than having people ask for the survey, I would suggest to include it in full as supplementary material

>> You have raised an important point however, we feel that it would be best to maintain that the survey be available only upon request, simply because the survey flow is quite complex. The survey was designed to have four major branches depending upon the nature of the participants’ relationship with the medical alert dog and it is therefore a long and complicated-looking document. If the editor and reviewer do not consider the length and flow of the survey to be an issue, we will be happy to provide the survey itself.

• Line 167: To make this sentence easier to read, I would add “for” between “knowledge,” and “what condition(s)”

>> This edit has been made on line 170. 

• Line 187: typo – “altered” should be “alerted”

>> This error has been corrected on line 190. 

Results

• Table 2: this is a very important table, that could be even more informative if it was also added how many dogs were trained/untrained when they alert to a particular condition. 

While we agree that this would be a very interesting addition to the table, unfortunately it is not possible to add this information. When participants with trained MADs were asked which condition their dog was first trained to alert to, they often also included the other conditions to which the dog began alerting in their response. Therefore, we are unable to differentiate between conditions for which the dog was trained and those that it was not trained for but began alerting. 

• Also, a table or figure should be able to stand on its own, meaning everything in it must be clear. Therefore, I would suggest explaining the abbreviations POTS and PTSD below the table.

>> Thank you for pointing out this error on our part. Table 2 has been edited accordingly and tables added since review have also included these footnotes (including the supplementary materials).

• Line 248: the numbers here (90% and 10%) do not correspond to those in table 5. Please check and adjust as necessary. 

>> The numbers you are referring to are now found in Table 3 and Table 4. In Table 3, we would like to bring your attention to the variable “Dog alerts to multiple conditions”. In this table, the total number of participants that reported that their dog does not alert to multiple conditions is 10. In Table 5, the total number of dogs that are reported to alert to single condition is also 10. Therefore, we are unable to identify any errors in the data between these tables. If we have incorrectly identified your specific area of concern, please do not hesitate to respond accordingly. 

• Lines 279-281: I would break this down for the most commonly alerted conditions as well.

>> We agree that this would be a valuable addition to the manuscript. We have therefore added Table 7, which presents the conditions to which dogs were reported to alert other people from most commonly reported to least commonly reported. We also conducted further analyses to examine whether or not dogs that alerted to these top six conditions were associated with whether or not a dog alerted to multiple people (Lines 285-289). Lastly, we also included further information in the supplementary material (S3 Table), which presents the data for the two dogs that were reported to alert other people to conditions other than those conditions they dog alerted to the target person. 

Discussion

• Line 316 and further: this discussion could be substantiated if the authors examined their data in a bit more detail for the most commonly reported conditions that dogs alert to (see major comment). At the moment it seems that dogs mainly alert the same condition in other people in the household, but surely that is more common for some conditions than others, since it is unlikely e.g. that two people from the same household have epilepsy. Although I do appreciate the conciseness with which the authors present their results, a more detailed presentation of the data would help here.

We appreciate the reviewer highlighting an important area where we could substantiate our claims. Additional analyses revealed no significant associations between whether dogs alerted anxiety and whether or not they alerted to other conditions, people, or both. But considering the conditions to which dogs were reported to alert to other people, further descriptive analyses revealed that anxiety was the second most common condition that dog alerted other people. We have further examined the most common conditions to which dogs alerted other people and have presented this data in Table 7. As indicated in the table, anxiety was the second most commonly reported condition that dogs alerted to other people. This point was discussed in the discussion at more length.

• Line 360: “for whether they decide to alert” – I would be careful with this kind of working. A dog does not decide to intentionally alert a person. The dog may decide to intentionally perform a particular behavior (which the owner then interprets as an alert).

>> Thank for you highlighting this point. We agree with your statement that dogs do not have a threshold for whether to decide to alert a person, but rather, have a threshold for whether they perform an alert behaviour. We have changed the wording of this sentence accordingly (line 384).

• Lines 476-477: also, how was the study presented to the respondents? Is there a risk of response bias in that respondents may have felt they were expected to have a dog that responds to multiple conditions/people. Perhaps the authors could elaborate on this as well?

>> Thank you for raising an important point. The information sheet presented to participants before completing the survey did in fact state that researchers were interested in whether or not their dog alerted to multiple conditions or multiple people. This could have resulted in demand characteristics influencing participants’ responses. This point has been discussed on lines 508-515.

We hope these changes are suitable. We feel that the revised manuscript is much stronger after making your recommended changes. Thank you for your time,

Dr Catherine Reeve

Animal Welfare and Behaviour

School of Psychology

Queen’s University Belfast

---

## [Decision Letter · Decision Letter 1]

25 Feb 2021

PONE-D-20-32952R1

Medical Alert Dogs are alerting to multiple conditions and multiple people

PLOS ONE

Dear Dr. Reeve,

Thank you for submitting your manuscript to PLOS ONE. After careful consideration, we feel that it has merit but does not fully meet PLOS ONE’s publication criteria as it currently stands. Therefore, we invite you to submit a revised version of the manuscript that addresses the points raised during the review process.

You will find the reviewer feedback as well as editorial feedback below.

We look forward to receiving your revised manuscript.

Kind regards,

I Anna S Olsson, Ph.D.

Academic Editor

PLOS ONE

Journal Requirements:

Additional Editor Comments (if provided):

Among the limitations of the study, please add a reflection on that the data are owner-reported, thus strictly speaking it is about *owners perceiving* their dogs to be alerting to multiple conditions. Given what you write on lines 510-511, "Given the results of other studies which found discrepancies between owner reports and objective assessments of MADs", it is important to mention this as a study limitation.

Reviewers' comments:

Reviewer's Responses to Questions

**Comments to the Author**

1. If the authors have adequately addressed your comments raised in a previous round of review and you feel that this manuscript is now acceptable for publication, you may indicate that here to bypass the “Comments to the Author” section, enter your conflict of interest statement in the “Confidential to Editor” section, and submit your "Accept" recommendation.

Reviewer #1: (No Response)

2. Is the manuscript technically sound, and do the data support the conclusions?

Reviewer #1: Yes

3. Has the statistical analysis been performed appropriately and rigorously? 

Reviewer #1: Yes

4. Have the authors made all data underlying the findings in their manuscript fully available?

Reviewer #1: Yes

5. Is the manuscript presented in an intelligible fashion and written in standard English?

Reviewer #1: Yes

6. Review Comments to the Author

Reviewer #1: Dear authors,

Thank you very much for the thorough consideration of all my comments. They were sufficiently addressed.

I have just two small remaining remarks (one that may require some revision still, the other is just a comment).

1) Since this survey was only done in English and distributed worldwide via the snowballing technique, how sure are you that the respondents indeed mastered the English language enough to understand your questions? Also, could cultural differences have an impact on the responses to your specific questions? It might be good to add a few lines about this in your discussion.

2) Regarding my previous comment about line 248 in the original document and numbers (10% and 90%) not matching with table 5: unfortunately I did not save a digital copy of your draft, nor did I retain the paper version where I made my first set of comments on. Editorial Manager also does not allow me to retrieve the original manuscript, so I cannot go back and check my own comment. Very sorry about this. But, looking at the current text and tables, it all seems clear now!

I look forward to seeing this paper as a published article!

7. PLOS authors have the option to publish the peer review history of their article (what does this mean?). If published, this will include your full peer review and any attached files.

Reviewer #1: No

---

## [Author Response · Author response to Decision Letter 1]

12 Mar 2021

Dear Dr Olsson,

Thank you and the reviewer for taking the time to review our manuscript. Thank you for taking the time to review our manuscript and have a reviewer provide feedback. We are pleased to have an opportunity to revise our manuscript once more. In the revised manuscript, we have carefully considered both you and the reviewer’s comments and suggestions and have addressed and/or responded to each of the points raised. The feedback was very helpful overall, and we are appreciative of such constructive feedback on our original submission. After addressing the issues raised, we feel the quality of the paper is much improved.

Please note that in our previous submission we neglected to address the comments/edit suggestion made by yourself, the editor. Therefore, we have addressed the previous comments as well as the most recent comments together here. 

Response to Editor Comments

Comment:

Response:

We have made formatting and style edits to the title page and manuscript according to the PLOs ONE style template.

Comment:

In your Methods section, please provide additional information about the participant recruitment method and the demographic details of your participants. Please ensure you have provided sufficient details to replicate the analyses such as: a) the recruitment date range (month and year), b) a description of any inclusion/exclusion criteria that were applied to participant recruitment, c) a table of relevant demographic details, d) a description of how participants were recruited, and e) descriptions of where participants were recruited and where the research took place.

Response:

A) Recruitment date range was added on Line 124.

B) Inclusion criteria were explicitly identified through the addition of a header on Line 128

C) Demographic details can be found in Table 1. 

D) Recruitment details were explicitly identified through the addition of a header on Line 121.

E) Where participants were recruited from was flagged by the reviewer in the previous round of revisions. This information has been added in Line 122. We also clarified that the survey was completed by participants on their own electronic device on Line 177.

Comment:

Please review your reference list to ensure that it is complete and correct. If you have cited papers that have been retracted, please include the rationale for doing so in the manuscript text or remove these references and replace them with relevant current references. Any changes to the reference list should be mentioned in the rebuttal letter that accompanies your revised manuscript. If you need to cite a retracted article, indicate the article’s retracted status in the References list and also include a citation and full reference for the retraction notice.

Response:

We have carefully reviewed all of our references and can confirm that they are accurate and that none have since been retracted. This is further confirmed by the fact that all papers have active DOIs (which have been added to the references as is highlighted in the revised manuscript)

Response to Reviewer Comments

Comment:

Among the limitations of the study, please add a reflection on that the data are owner-reported, thus strictly speaking it is about owners perceiving their dogs to be alerting to multiple conditions. Given what you write on lines 510-511, "Given the results of other studies which found discrepancies between owner reports and objective assessments of MADs", it is important to mention this as a study limitation.

Response:

Thank you for identifying this important point. We have added Lines 509-513 which highlight this aspect of the study and caution the reader to interpret the results accordingly.

Comment:

Since this survey was only done in English and distributed worldwide via the snowballing technique, how sure are you that the respondents indeed mastered the English language enough to understand your questions? Also, could cultural differences have an impact on the responses to your specific questions? It might be good to add a few lines about this in your discussion.

Response:

Regarding the first point, we did not require that participants indicate that they had mastered the English language at any point in the survey. We did require, however, that participants read the participant information sheet and then consent to a series of statement prior to beginning the survey. We presume that anyone who could not comprehend the information sheet and the consent statements would not have consented to participating.

We do not have any reason to believe that participants’ responses to the questions themselves could be affected by cultural differences. We did, however, add a comment about the potential for participants’ interpretation of their dogs’ behaviour to be different across cultures (Line 511), as supported by the addition of another reference (64, Amici et al., 2019).

We hope these changes are suitable. We feel that the revised manuscript is much stronger after making your recommended changes. Thank you for your time,

Dr Catherine Reeve

Animal Welfare and Behaviour

School of Psychology

Queen’s University Belfast

---

## [Editor Report · Decision Letter 2]

15 Mar 2021

Medical Alert Dogs are alerting to multiple conditions and multiple people

PONE-D-20-32952R2

Dear Dr. Reeve,

We’re pleased to inform you that your manuscript has been judged scientifically suitable for publication and will be formally accepted for publication once it meets all outstanding technical requirements.

Kind regards,

I Anna S Olsson, Ph.D.

Academic Editor

PLOS ONE
---

## [Editor Report · Acceptance letter]

22 Mar 2021

PONE-D-20-32952R2 

Medical Alert Dogs are alerting to multiple conditions and multiple people 

Dear Dr. Reeve:

I'm pleased to inform you that your manuscript has been deemed suitable for publication in PLOS ONE. Congratulations! Your manuscript is now with our production department. 

Kind regards, 

on behalf of

Dr. I Anna S Olsson 

Academic Editor

PLOS ONE